# InfiMed-Foundation: Pioneering Advanced Multimodal Medical Models with Compute-Efficient Pre-Training and Multi-Stage Fine-Tuning

## Abstract

Multimodal large language models (MLLMs) have shown remarkable potential in various domains, yet their application in the medical field is hindered by several challenges. General-purpose MLLMs often lack the specialized knowledge required for medical tasks, leading to uncertain or hallucinatory responses. Knowledge distillation from advanced models struggles to capture domain-specific expertise in radiology and pharmacology. Additionally, the computational cost of continual pretraining with large-scale medical data poses significant efficiency challenges. To address these issues, we propose InfiMed-Foundation-1.7B and InfiMed-Foundation-4B, two medical-specific MLLMs designed to deliver state-of-the-art performance in medical applications. We combined high-quality general-purpose and medical multimodal data and proposed a novel five-dimensional quality assessment framework to curate high-quality multimodal medical datasets. We employ low-to-high image resolution and multimodal sequence packing to enhance training efficiency, enabling the integration of extensive medical data. Furthermore, a three-stage supervised fine-tuning process ensures effective knowledge extraction for complex medical tasks. Evaluated on the MedEvalKit framework, InfiMed-Foundation-1.7B outperforms Qwen2.5VL-3B, while InfiMed-Foundation-4B surpasses HuatuoGPT-V-7B and MedGemma-27B-IT, demonstrating superior performance in medical visual question answering and diagnostic tasks. By addressing key challenges in data quality, training efficiency, and domain-specific knowledge extraction, our work paves the way for more reliable and effective AI-driven solutions in healthcare.

## 1 Introduction

In recent years, multimodal large language models (MLLMs) have demonstrated remarkable capabilities across various domains (Hurst et al., 2024; Team et al., 2025a; Zhu et al., 2025), achieving near-expert-level performance in tasks such as visual question answering (VQA), image captioning, and text generation. However, general-purpose MLLMs often lack the specialized knowledge required to address domain-specific challenges, particularly in the medical field (Lee et al., 2023). When tasked with medical queries, these models frequently produce uncertain or even hallucinatory responses (Li et al., 2023; Chen et al., 2024b), highlighting the need for domain-specific adaptations. The medical domain demands a high level of precision, reliability, and domain expertise, as inaccurate outputs can have significant consequences in clinical settings.

Recent efforts have focused on integrating medical multimodal data with general-purpose MLLMs to develop medical-specific models (Hyland et al., 2023; Team et al., 2025b). For instance, Lingshu (Team et al., 2025b) utilized a diverse set of open-source medical data, general-purpose data, and high-quality synthetic medical data to train a model that achieved promising results across various medical evaluation benchmarks. These advancements underscore the potential of tailored MLLMs in tasks such as medical VQA and report generation. Despite these achievements, existing medical MLLMs face several challenges that limit their effectiveness and scalability. Firstly, many medical MLLMs rely on knowledge distillation from advanced general-purpose models (Hurst et al., 2024;

Jaech et al., 2024) to curate training data. While effective in some contexts, this approach struggles to capture the extensive domain-specific expertise required for fields such as radiology, pharmacology, and pathology. Secondly, the absence of supervision by medical professionals during the distillation process consequently elevates the risk of generating model hallucinations. Thirdly, to inject comprehensive medical knowledge through continual pretraining, large-scale high-quality medical data is essential. However, processing such data is computationally expensive, necessitating strategies to enhance pretraining efficiency.

In this work, we propose InfiMed-Foundation-1.7B and InfiMed-Foundation-4B, two medical-specific MLLMs that achieve state-of-the-art performance across multiple medical benchmarks. To address the aforementioned challenges, we curated a high-quality multimodal medical dataset, combining carefully selected medical data with general-purpose multimodal data. In collaboration with medical professionals, we developed a novel five-dimensional quality assessment framework to ensure the reliability and relevance of the training data. During continual pretraining, we optimized computational efficiency by reducing the number of image patches to 144 and introducing multimodal sequence packing, which allowed us to incorporate a larger volume of medical data. Furthermore, we designed a three-stage supervised fine-tuning (SFT) process, comprising general instruction following, medical instruction following, and cross-distribution instruction adaptation. This structured approach enables our models to progressively acquire the ability to address complex medical tasks effectively.

We evaluated our models using the MedEvalKit framework (Team et al., 2025b), a comprehensive suite of medical benchmarks. Experimental results demonstrate that InfiMed-Foundation-1.7B outperforms the Qwen2.5VL-3B model (Bai et al., 2025), while InfiMed-Foundation-4B surpasses both the HuatuoGPT-V-7B (Chen et al., 2024b) and MedGemma-27B-IT (Sellergren et al., 2025) models. Through ablation studies, we validated the critical role of our multi-stage SFT strategy. Additionally, case studies in medical VQA and diagnostic tasks highlight the superior performance of our models, showcasing their potential to assist clinicians in real-world scenarios.

Our contributions can be summarized as follows:

- **Data Curation**: We introduce a five-dimensional quality assessment framework, developed in collaboration with medical professionals, to select high-quality medical datasets, ensuring robustness and reliability in training.
- **Training Efficiency**: We enhance pretraining efficiency by adopting multimodal sequence packing and reducing image patch counts, enabling the incorporation of extensive medical data while minimizing computational costs.
- **Performance**: Our InfiMed-Foundation models achieve outstanding results across multiple medical evaluation benchmarks, setting a new standard for medical-specific MLLMs.

## 2 RELATED WORK

**Medical-Specific Multimodal Models** Medical-specific MLLMs have gained traction for tasks such as clinical reasoning, medical VQA, and report generation. LLaVA-Med (Li et al., 2023) pioneered this domain by utilizing large-scale biomedical image-text pairs from PubMed Central for concept alignment, enabling the model to learn domain-specific visual vocabulary, followed by instruction tuning with GPT-4-generated data. However, the low quality of PubMed data often leads to hallucinations and weak reasoning capabilities, as the instruction data relies solely on textual captions and contexts without leveraging biomedical images. In contrast, HuatuoGPT-Vision (Chen et al., 2024b) employs GPT-4V, a multimodal model capable of processing both images and text, to denoise PubMed data and create the high-quality PubMedVision dataset for SFT, improving performance in medical VQA. MedGemma (Sellergren et al., 2025) adopts a multi-stage training pipeline. First, the vision encoder is enhanced using medical image-text pairs. Subsequently, the language model undergoes continual pretraining and is then re-adapted with the vision encoder. Finally, the model is refined through distillation and reinforcement learning. Despite these advances, existing models often lack robust data quality control and professional supervision, limiting their reliability in clinical settings. Our InfiMed-Foundation models address these issues by combining high-quality general and medical multimodal data. A novel five-dimensional quality assessment framework, developed with medical professionals, ensures robust performance.

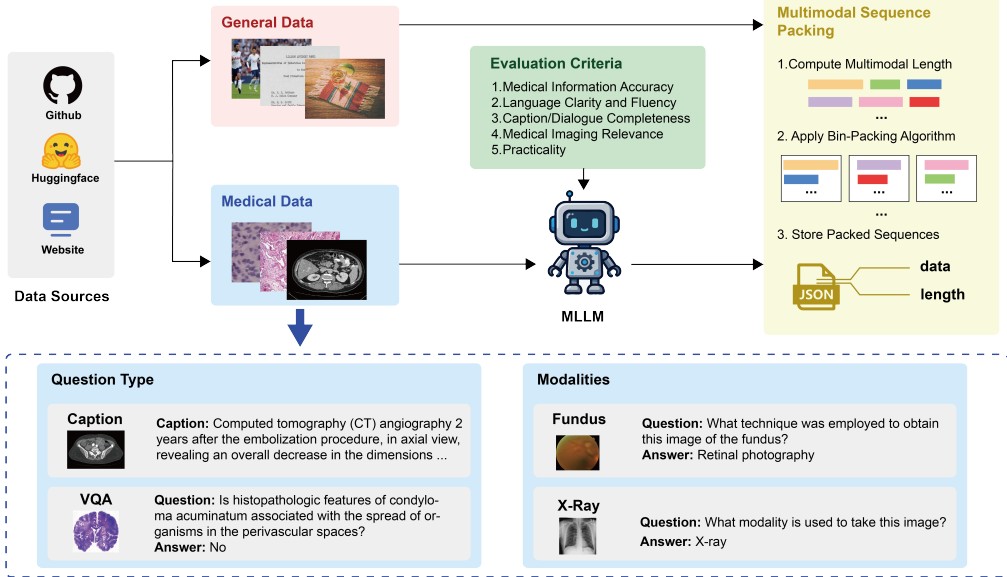

Figure 1: The pipeline of our dataset curation.

**Efficient Training Strategies for Multimodal Models** MLLM pretraining aims to align different modalities in a shared embedding space, requiring large-scale image-text pair datasets. Many approaches train only the vision-language projector to mitigate catastrophic forgetting and reduce computational costs while freezing the vision encoder and language model (Jin et al., 2024). However, TinyLLaVA (Zhou et al., 2024) notes that training only the projector may lead to suboptimal alignment in small-scale LLMs, proposing partial unfreezing of the vision encoder and LLM to improve modality alignment. Similarly, VILA (Lin et al., 2024) demonstrates that unfreezing LLM parameters during pretraining is essential for inheriting in-context learning capabilities, which are critical for personalized medical recommendations in clinical settings. To further enhance efficiency, Idefics2 (Laurençon et al., 2024) employs a perceiver resampler to reduce visual token counts. It adopts a two-stage pretraining approach, using lower image resolutions in the initial stage to accelerate basic alignment. Inspired by these works, our InfiMed-Foundation models unfreeze both the projector and LLM parameters during pretraining and introduce multimodal sequence packing with a reduced image patch count of 144, significantly improving computational efficiency while enabling the integration of extensive medical data.

## 3 DATASET CURATION

To train our proposed InfiMed-Foundation series models, we curated a large-scale, heterogeneous dataset comprising both medical and general-domain multimodal data. The pipeline of our dataset curation is shown in Figure 1. The medical dataset covers diverse modalities, body parts, question types, and multiple languages. To improve the model's generalization and linguistic capabilities, we further curated a large-scale general-domain dataset encompassing diverse real-world scenarios. To ensure data quality, we developed an evaluation pipeline based on LLMs, which was used to assess and filter the collected datasets. An overview of all collected datasets is provided in Table 1.

### 3.1 DATA COLLECTION

**Medical Data** To build a high-quality medical dataset for training our InfiMed-Foundation models, we collected and aggregated a range of multimodal medical datasets from public sources, which include image-text pairs. Moreover, we divided them into two categories: caption and instruction. The collected multimodal medical data span various modalities (e.g., pathology, microscopy, and CT), body parts (e.g., head, neck, and chest), question types (e.g., open-ended, closed-ended, and multiple-choice), and multiple languages (e.g., English and Chinese).

**General Data** To enable the multimodal large language model to achieve strong multimodal understanding and visual reasoning capabilities, it is necessary to conduct continual pretraining on large-

Table 1: Overview of Training datasets.

| Type of Data | Collected Datasets |
| --- | --- |
| General Caption Data | DataComp (Gadre et al., 2023), CCS (Li et al., 2022) |
| General Interleaved Data | OBELICS (Laurençon et al., 2023), mmc4 (Zhu et al., 2023) |
| General Instruction Data | Mammoth-VL (Guo et al., 2024) |
| Medical Caption Data | IU-Xray (Chen et al., 2020), LLaVA-Med (Li et al., 2023), LLaVA-Med-60K-IM-text (Kang, 2024), Medtrinity-25M (Xie et al., 2025), MedPix-2.0 (Siragusa et al., 2025), PMC-OA (Lin et al., 2023), PubMedVision (Chen et al., 2024a), ROCO (Pelka et al., 2018), ROCOv2 (Rückert et al., 2024) |
| Medical Instruction Data | LLaVA-Med (Li et al., 2023), Path-VQA (He et al., 2020), PMC-VQA (Zhang et al., 2023), PubMedVision (Chen et al., 2024a), SLAKE (Liu et al., 2021), VQA-Med-2019 (Ben Abacha et al., 2019), VQA-RAD (Lau et al., 2018) |

scale image-text caption datasets and interleaved datasets. For general data collection, we followed the highly open-source Open-Qwen2VL (Wang et al., 2025). Specifically, for general caption data, we used two subsets of DataComp-Medium-128M (Gadre et al., 2023), filtered by Data-Filtering-Network (DFN) (Fang et al., 2023) and MLM-Filter (Wang et al., 2024), respectively. Additionally, we incorporated high-quality caption data from the BLIP (Li et al., 2022), which was filtered from a combination of three web datasets: CC3M (Changpinyo et al., 2021), CC12M (Changpinyo et al., 2021), and SBU (Ordonez et al., 2011) (CCS). For general interleaved data, we employed high-quality subsets of the OBELICS dataset (Laurençon et al., 2023) and the MMC4 dataset (Zhu et al., 2023) to enhance the multimodal in-context learning ability. And we utilized the MAmmoTH-VL-10M (Guo et al., 2024) to bolster the model's instruction-following and reasoning capabilities.

## 3.2 DATA EVALUATION

We performed quality control on medical data using both LLM-based and manual inspection. To evaluate data quality, we randomly sampled 500 samples from each dataset and conducted a detailed evaluation. We collaborated with a group of medical professionals to define five evaluation criteria:

1. **Medical Information Accuracy**: Assess how medically accurate and clinically appropriate the information in the sample is.
2. **Language Clarity and Fluency**: Assess how well the content is communicated in natural, readable, and professional language.
3. **Caption/Dialogue Completeness**: Access whether the caption/dialogue directly, sufficiently, and contextually addresses the input question or medical concern.
4. **Medical Imaging Relevance**: For samples that include an image, assess whether the image clearly supports or corresponds to the associated text.
5. **Practicality**: Assess how useful the data sample is for real-world medical applications, such as clinical decision support or patient communication.

Each dimension was rated on a scale of 1 to 5. Detailed scoring guidelines are provided in Appendix A.2. We employed GPT-o3 as an automated evaluator to rate the sampled data according to the above criteria. The results guided our filtering process and informed our overall data quality assessment. After our quality assessment, we excluded some datasets, including IU-Xray, MedPix-2.0, PMC-OA, and VQA-Med-2019.

## 3.3 MULTIMODAL SEQUENCE PACKING

Owing to the variable sequence lengths inherent in multimodal data, direct training often necessitates padding all samples to a uniform maximum length. This practice introduces a substantial number of padding tokens, resulting in significant computational inefficiency. To address this issue, we employ a multimodal sequence-packing strategy during continuous pretraining. This method reorganizes

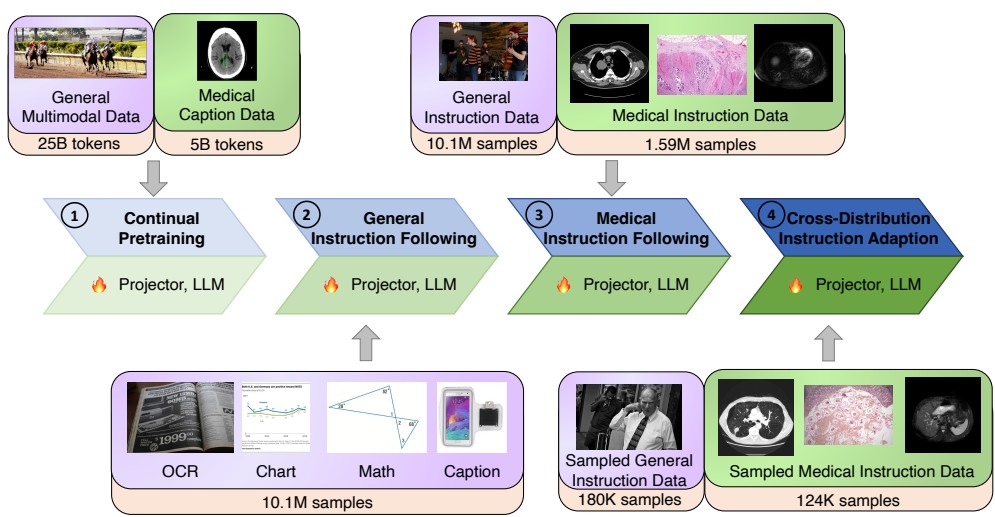

Figure 2: The training pipeline of InfiMed-Foundation models, which consists of four stages: continual pretraining, general instruction following, medical instruction following, and cross-distribution instruction adaption.

multiple multimodal samples into consolidated sequences with total lengths approaching the model's maximum context window of 4,096 tokens.

The multimodal sequence packing procedure consists of the following three steps:

1. **Compute Multimodal Length:** The token length of each multimodal sample, which incorporates both visual and textual elements, is calculated.
2. **Apply Bin-Packing Algorithm:** Samples are sorted in descending order of their lengths and subsequently packed into bins using the First-Fit-Decreasing (FFD) bin packing algorithm (Johnson, 1973). The objective is to aggregate samples into bins such that the cumulative length of each bin is as close as possible to, but does not exceed, 4,096 tokens.
3. **Store Packed Sequences:** Each consolidated bin of packed sequences is saved into a JSON file. The structure of the file is organized as a dictionary containing the following two key fields:
   - `"data"`: A list of the regrouped samples. Each sample within the list is a dictionary itself, containing the Base64-encoded image data, the corresponding text, and other metadata.
   - `"lengths"`: A list of integers that records the original multimodal sequence length of each constituent sample within the bin.

## 4 MODEL TRAINING

InfiMed-Foundation models consist of three key components: a LLM, a vision encoder, and a lightweight MLP-based visual projector. We adopt this architecture with Qwen3-Instruct series LLMs (Yang et al., 2025), SigLIP-SO-400M Vision Encoder (Tang et al., 2002), and Adaptive Average-Pooling Visual Projector (Yao et al., 2024). Specifically, the visual projector consists of an Adaptive Average-Pooling layer followed by a two-layer MLP. The adaptive pooling layer allows us to flexibly rescale the fixed output of 729 visual patches from the SigLIP encoder to any desired number of visual tokens. During pretraining, we downsample the visual representation to 144 visual tokens per image to reduce computational cost and encourage global abstraction. In the supervised fine-tuning (SFT) stage, we revert to the full 729 patches resolution to capture more detailed visual cues. This design offers a good trade-off between efficiency and flexibility, allowing the model to adaptively balance global and local visual features across different training phases.

### 4.1 TRAINING RECIPE

During both the pretraining and supervised fine-tuning (SFT) stages, we freeze the parameters of the vision encoder to reduce computational cost. Only the LLM and the projector are trainable.

In the pretraining stage, we train the model on a large corpus comprising general-domain multimodal samples ($\sim$25B tokens) and medical samples ($\sim$5B tokens). And inspired by Allen-Zhu & Li (2024), we adopt Instruct series models instead of base series models for pretraining, aiming to enhance the model's contextual understanding and adaptation efficiency in the subsequent alignment stages.

Inspired by the multi-stage alignment strategy proposed in Lingshu (Team et al., 2025b), we design a three-stage SFT pipeline to progressively inject and align different capabilities into the model as illustrated in Figure 2. The three stages in our pipeline are: 1) General Instruction Following: Enhancing the model's ability to follow general-purpose instructions in diverse contexts. 2) Medical Instruction Following: Fine-tuning the model for medical-domain tasks that require domain-specific reasoning and understanding. 3) Cross-distribution Instruction Adaptation: To ensure generalization across heterogeneous data sources, we apply down-sampling to each dataset, enforcing inter-dataset balance. This prevents overfitting to high-resource instruction types and encourages the model to adapt to a wide range of data distributions. The details of the data mixture for different training stages are provided in Appendix A.3.

### 4.1.1 COMPUTE-EFFICIENT PRETRAINING

The primary objective of pretraining multimodal medical models is to achieve robust alignment between image and text modalities while injecting domain-specific medical knowledge into the model. This alignment enhances the model's ability to understand medical visual data, laying a critical foundation for subsequent knowledge extraction in tasks such as medical VQA and diagnostic support (Li et al., 2023; Chen et al., 2024b). However, pretraining large-scale multimodal models, especially with high-resolution medical images and diverse text data, is computationally intensive.

In our pretraining phase, we freeze the vision encoder to preserve its pretrained feature extraction capabilities, while updating the parameters of the LLM and the projector. Our pretraining dataset comprises high-quality general-purpose multimodal data and medical image-text pairs curated using our five-dimensional quality assessment framework. To enhance the efficiency of pretraining, we employ multimodal sequence packing, a strategy that concatenates multimodal data of varying lengths into sequences approaching a maximum length of 4096 tokens. This approach maximizes computational resource utilization by minimizing padding and ensuring dense data processing. Additionally, we implement adaptive average-pooling to reduce the number of tokens representing images to 144. This reduction mitigates computational overhead while preserving essential visual features.

### 4.1.2 GENERAL INSTRUCTION FOLLOWING

The first stage of our SFT pipeline, termed General Instruction Following, aims to endow the InfiMed-Foundation models with robust multimodal reasoning and instruction-following capabilities, establishing a strong foundation for subsequent medical domain adaptation. Unlike medical-specific fine-tuning, which focuses on domain knowledge, this stage emphasizes general multimodal understanding, ensuring the model can handle complex reasoning tasks before specializing in medical applications. We leverage the MAmmoTH-VL-10M dataset (Guo et al., 2024), a large-scale multimodal instruction-tuning dataset designed to foster reasoning-intensive capabilities.

The MAmmoTH-VL-10M dataset is specifically curated to address the limitations of traditional instruction datasets, which often focus on simple VQA tasks with phrase-based responses lacking detailed reasoning processes. To construct MAmmoTH-VL-10M, data sources underwent manual screening to categorize based on the information density of responses. Subsequently, a combination of MLLMs and LLMs was used to rewrite responses, generating detailed rationales. Finally, an MLLM-based filtering step ensured logical consistency and reliability of the rationales. By training on MAmmoTH-VL-10M's detailed rationales, our models develop enhanced reasoning abilities, which are critical for subsequent medical-specific fine-tuning stages.

### 4.1.3 MEDICAL INSTRUCTION FOLLOWING

To equip the model with medical question-answering capabilities, the second stage of our fine-tuning pipeline is dedicated to medical instruction tuning. In this stage, the model is trained on a collection of high-quality medical VQA datasets that have passed the rigorous data quality assessment detailed

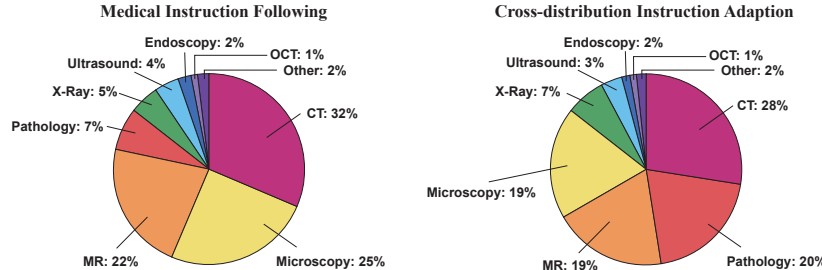

Figure 3: The modality distribution of the dataset in the Medical Instruction Following stage and the Cross-distribution Instruction Adaptation stage.

in Section 3.2. These datasets span a diverse range of medical subdomains, including clinical diagnostics, radiological interpretation, and pathological analysis, each requiring varying degrees of domain-specific reasoning and precision.

To avoid catastrophic forgetting of general capabilities acquired in the first stage, we incorporate general instruction-following data into this phase. This joint training strategy allows the model to retain its general multimodal abilities while enhancing its understanding of medical knowledge. Importantly, this approach prevents the model from overfitting to narrow medical domains and promotes better generalization across both general-domain and medical-domain tasks.

### 4.1.4 CROSS-DISTRIBUTION INSTRUCTION ADAPTION

Naively fine-tuning on a mixture of medical VQA datasets in a single stage risks overfitting to high-resource datasets, as they disproportionately influence the training objective. This imbalance undermines the model's generalization to low-resource tasks and diverse instruction distributions. To address this, the third stage introduces cross-distribution instruction adaptation. In this stage, we construct mixed instruction datasets by sampling from multiple datasets across both general and medical domains. Furthermore, we also balance the number of samples between medical and general instruction datasets to prevent distributional bias during training. To ensure inter-dataset balance, we down-sample large datasets to bring all sources to a similar scale, avoiding domination by high-resource datasets. The modality distribution of the dataset in the Medical Instruction Following stage and the Cross-distribution Instruction Adaptation stage is shown in Figure 3.

Importantly, this stage retains the same architectural configuration and optimization settings as previous stages. This approach preserves training stability while enhancing the model's ability to generalize across heterogeneous datasets and instruction formats.

## 5 EXPERIMENTS

To comprehensively evaluate our model, InfiMed-Foundation, we compare its performance against a diverse set of baseline models on various medical benchmarks. These baselines include proprietary and open-source models, with the latter encompassing general-purpose and medical-specific models. Specifically, our evaluation includes the following models: **Proprietary Models**: GPT-5, GPT-5-mini, GPT-5-nano, GPT-4.1 (OpenAI, 2025), Claude Sonnet 4 (Anthropic, 2025), Gemini-2.5-Flash (Comanici et al., 2025). **General Open-source Models**: Qwen2.5-VL-Instruct (Bai et al., 2025), InternVL3 (Zhu et al., 2025). **Medical Open-source Models**: MedGemma (Sellergren et al., 2025), LLaVA-Med (Li et al., 2023), HuatuoGPT-V (Chen et al., 2024b), Lingshu (Team et al., 2025b), BioMediX2 (Mullappilly et al., 2024).

To ensure a fair comparison, all models are evaluated using MedEvalKit (Team et al., 2025b), a systematic evaluation framework. This framework assesses performance across mainstream medical benchmarks, including multiple-choice questions, open-ended questions, and other task formats.

### 5.1 EVALUATION BENCHMARK

To comprehensively evaluate the performance of medical MLLMs, we utilized a diverse set of medical benchmark datasets. These benchmarks include multiple-choice, open-ended, and closed-ended questions, covering datasets such as the Health & Medical subset of MMMU (Yue et al.,

Table 2: Results of comparison of InfiMed with other MLLMs on medical multimodal benchmarks. Note that OMVQA and MedXQA indicate OmniMedVQA and MedXpertQA-Multimodal benchmarks, respectively. Models colored in gray denote our InfiMed.

| Model | MMMU-Med | VQA-RAD | SLAKE | PathVQA | PMC-VQA | OMVQA | MedXVQA | Avg. |
|---|---|---|---|---|---|---|---|---|
| **Proprietary Models** | | | | | | | | |
| GPT-5 | 83.6 | 67.8 | 78.1 | 52.8 | 60.0 | 76.4 | 71.0 | 70.0 |
| GPT-5-mini | 80.5 | 66.3 | 76.1 | 52.4 | 57.6 | 70.9 | 60.1 | 66.3 |
| GPT-5-nano | 74.1 | 55.4 | 69.3 | 45.4 | 51.3 | 66.5 | 45.1 | 58.2 |
| GPT-4.1 | 75.2 | 65.0 | 72.2 | 55.5 | 55.2 | 75.5 | 45.2 | 63.4 |
| Claude Sonnet 4 | 74.6 | 67.6 | 70.6 | 54.2 | 54.4 | 65.5 | 43.3 | 61.5 |
| Gemini-2.5-Flash | 76.9 | 68.5 | 75.8 | 55.4 | 55.4 | 71.0 | 52.8 | 65.1 |
| **General Open-source Models** | | | | | | | | |
| Qwen2.5VL-3B | 51.3 | 56.8 | 63.2 | 37.1 | 50.6 | 64.5 | 20.7 | 49.2 |
| Qwen2.5VL-7B | 50.6 | 64.5 | 67.2 | 44.1 | 51.9 | 63.6 | 22.3 | 52.0 |
| InternVL3-8B | 59.2 | 65.4 | 72.8 | 48.6 | 53.8 | 79.1 | 22.4 | 57.3 |
| **Medical Open-source Models** | | | | | | | | |
| MedGemma-4B-IT | 43.7 | 49.9 | 76.4 | 48.8 | 49.9 | 69.8 | 22.3 | 51.5 |
| LLaVA-Med-7B | 29.3 | 53.7 | 48.0 | 38.8 | 30.5 | 44.3 | 20.3 | 37.8 |
| HuatuoGPT-V-7B | 47.3 | 67.0 | 67.8 | 48.0 | 53.3 | 74.2 | 21.6 | 54.2 |
| Lingshu-7B | 54.0 | 67.9 | 83.1 | 61.9 | 56.3 | 82.9 | 26.7 | 61.8 |
| BioMediX2-8B | 39.8 | 49.2 | 57.7 | 37.0 | 43.5 | 63.3 | 21.8 | 44.6 |
| MedGemma-27B-IT | 56.2 | 62.3 | 74.9 | 44.4 | 49.5 | 66.3 | 33.9 | 55.4 |
| InfiMed-Foundation-1.7B | 34.7 | 56.3 | 75.3 | 60.7 | 48.1 | 58.9 | 21.8 | 50.8 |
| InfiMed-Foundation-4B | 43.3 | 57.9 | 77.7 | 63.4 | 56.6 | 76.8 | 21.9 | 56.4 |

2024), VQA-RAD (Lau et al., 2018), SLAKE (Liu et al., 2021), PathVQA (He et al., 2020), PMC-VQA (Zhang et al., 2023), the open-source portion of OmniMedVQA (Hu et al., 2024), and the multimodal subset of MedXpertQA (Zuo et al., 2025). These datasets encompass various medical imaging modalities, including CT scans, dermoscopy, X-rays, and microscopy images. This variety enables a robust assessment of the model's ability to process and interpret diverse medical visual data, ensuring a comprehensive evaluation of its medical reasoning and multimodal understanding capabilities. Our evaluation framework and implementation details can be found in the Appendix A.4 and Appendix A.5.

## 5.2 MAIN RESULTS

Table 2 presents a comprehensive comparison of our proposed models, InfiMed-Foundation-1.7B and InfiMed-Foundation-4B, against both proprietary and open-source MLLMs across seven representative medical benchmarks. The final column shows the macro-average across all benchmarks.

InfiMed-Foundation-4B achieves an average accuracy of 56.4%, outperforming all general and medical open-source MLLMs of comparable scale and closing the gap with several strong proprietary systems. Notably, InfiMed-Foundation-4B exceeds MedGemma-27B-IT (+1.0%) and HuatuoGPT-V-7B (+2.2%) despite having fewer parameters, demonstrating superior parameter efficiency. Compared to LLaVA-Med-7B, a widely-used baseline, InfiMed-Foundation-4B shows a substantial +18.6% gain on average performance. Among individual benchmarks, InfiMed-Foundation-4B achieves particularly strong results on PathVQA (63.4%), outperforming all open-source medical baselines, and on SLAKE (77.7%), where it ranks second only to Lingshu-7B (83.1%). While performance on MedXVQA remains modest (21.9%), this is consistent with trends across other open models and highlights the dataset's unique challenges. Interestingly, InfiMed-Foundation-1.7B maintains competitive performance across most datasets despite its smaller scale, indicating that our architecture and training approach are robust across sizes. For instance, it surpasses BioMediX2-8B by +6.2% on average, despite having one-fifth the parameter count. We include a set of comparative case studies of our InfiMed-Foundation-4B model versus Qwen2.5-VL-7B in the Appendix A.6.

While proprietary models like GPT-5 (70.0%) and Gemini-2.5-Flash (65.1%) still lead in overall accuracy, our results demonstrate that InfiMed-Foundation-4B achieves state-of-the-art performance among open-source medical MLLMs, and narrows the performance gap with closed models significantly, especially considering compute and scale limitations.

Table 3: Ablation study results for SFT stages on medical multimodal benchmarks.

| SFT Stage | | | Medical Benchmarks | | | | | | | |
|---|---|---|---|---|---|---|---|---|---|---|
| Stage 1 | Stage 2 | Stage 3 | MMMU-Med | VQA-RAD | SLAKE | PathVQA | PMC-VQA | OMVQA | MedXVQA | Avg. |
| ✓ | | | 42.7 | 53.7 | 62.3 | 51.6 | 54.2 | 71.8 | 21.9 | 51.2 |
| ✓ | ✓ | | 41.3 | 53.0 | 73.1 | 50.5 | 58.8 | 77.8 | 19.0 | 53.4 |
| ✓ | | ✓ | 41.3 | 53.4 | 76.0 | 61.7 | 51.0 | 72.0 | 21.3 | 53.8 |
| ✓ | ✓ | ✓ | 43.3 | 57.9 | 77.7 | 63.4 | 56.6 | 76.8 | 21.9 | 56.4 |

## 5.3 ABLATION STUDY

To investigate the contributions of each stage in the SFT process outlined in Section 4.1, we conducted an ablation study by selectively applying the three SFT stages: General Instruction Following (Stage 1), Medical Instruction Following (Stage 2), and Cross-distribution Instruction Adaptation (Stage 3). This study aims to quantify the impact of each stage on the performance of our multimodal medical large language model across various benchmarks. The results are summarized in Table 3.

Each SFT stage contributes differently to the model's performance across various tasks. When omitting Stage 2 (Medical Instruction Following) and performing only Stage 1 (General Instruction Following), we observe a significant performance drop on benchmarks such as SLAKE, PMC-VQA, and OmniMedVQA. This underscores the importance of incorporating high-quality medical VQA datasets during Stage 2, which enhances the model's ability to address domain-specific medical queries effectively.

In contrast, the inclusion of Stage 3 (Cross-distribution Instruction Adaptation) leads to substantial performance improvements, particularly on VQA-RAD, SLAKE, and PathVQA, with accuracy gains of 4.9%, 4.6%, and 12.9%, respectively, over the configuration with only Stages 1 and 2. This indicates that Stage 3 effectively mitigates the risk of model domination by larger medical datasets, enabling better generalization across diverse data distributions. By adapting the model to handle cross-distribution variations, Stage 3 ensures robust performance on benchmarks with differing data characteristics, such as the radiologically focused VQA-RAD and the pathology-oriented PathVQA. Furthermore, when performing only Stages 1 and 3, we observe improved performance on SLAKE and PathVQA compared to the configuration with only Stages 1 and 2, with accuracy gains of 3.1%, and 11.2%, respectively. This improvement is attributed to Stage 3's ability to mitigate the risk of model domination by larger medical datasets. However, this configuration results in a notable performance drop on PMC-VQA and OmniMedVQA, with accuracy reductions of 7.8% and 5.8%, respectively. These results highlight the necessity of including Stage 2 to leverage more medical-related data for maintaining robust performance. Therefore, the optimal configuration requires all three stages.

The ablation study demonstrates the complementary nature of the three stages: Stage 1 establishes a strong foundation in general instruction following, Stage 2 enhances medical domain expertise, and Stage 3 ensures adaptability to diverse data distributions. Together, these stages enable our model to achieve state-of-the-art performance in multimodal medical tasks.

## 6 CONCLUSION

In this work, we introduce InfiMed-Foundation-1.7B and InfiMed-Foundation-4B, two medical-specific multimodal large language models. We present a novel five-dimensional quality assessment framework developed with medical professionals to obtain a curated high-quality multimodal medical dataset. By optimizing pretraining efficiency with multimodal sequence packing and scaling down image patches, we incorporated extensive medical data cost-effectively. Our three-stage supervised fine-tuning process enabled robust performance across complex medical tasks. Evaluations using the MedEvalKit framework showed that InfiMed-Foundation-1.7B outperforms Qwen2.5VL-3B, while InfiMed-Foundation-4B surpasses HuatuoGPT-V-7B and MedGemma-27B-IT, setting new standards for medical MLLMs. Ablation studies and case studies in medical VQA and diagnostics confirmed the critical role of our SFT strategy and data curation, highlighting the models' potential to assist clinicians. Our contributions in data curation, training efficiency, and performance pave the way for scalable medical AI, with future work aimed at optimizing the vision encoder and expanding data diversity to further enhance model capabilities.

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

# A APPENDIX

## A.1 THE USE OF LARGE LANGUAGE MODELS

We used large language models (LLMs) to check for grammatical inaccuracies and to improve the clarity and flow of the text. By helping to articulate the presented ideas more precisely, the use of LLMs contributed to enhancing the document's readability.

## A.2 SCORING GUIDELINES

This appendix provides the detailed scoring guidelines used to evaluate the quality of the sampled data across five dimensions: (1) Medical Information Accuracy, (2) Language Clarity and Fluency, (3) Dialogue Completeness, (4) Medical Imaging Relevance, and (5) Practicality. Each dimension is scored on a 1–5 scale, with higher scores indicating better quality. The guidelines were designed in collaboration with medical experts to ensure domain relevance and consistency.

---

**Prompt for Quality Assessment**

You are evaluating the quality of a single data sample from medical datasets, including three types: visual question answering, captioning, and case reporting. Rate the sample on a scale of 1 to 5 for each of the following five dimensions, and provide a clear explanation for your score. Your response must be in a valid JSON format, strictly following the structure below.

**Evaluation Dimensions and Guidelines**

**1. Medical Information Accuracy:** Definition: How medically accurate and clinically appropriate is the information in this sample? Evaluate whether the diagnosis, symptoms, treatment, terminology, and reasoning are factually correct and aligned with standard medical knowledge.

- 1 – Contains serious factual errors or misinformation; could lead to harm.
- 2 – Includes noticeable inaccuracies or misconceptions; questionable clinical logic.
- 3 – Mostly accurate, but includes some outdated, vague, or imprecise information.
- 4 - Clinically sound and reliable, with only minor wording or factual issues.
- 5 - Fully medically accurate, consistent with guidelines and expert-level clarity.

**2. Language Clarity and Fluency:** Definition: How well is the information communicated in natural, readable, and professional language? Assess grammar, clarity, flow, and appropriateness for medical or patient-facing communication.

- 1 – Unclear or disorganized; major grammar issues that hinder understanding.
- 2 – Awkward, ambiguous, or frequently incorrect language.
- 3 – Understandable but with some unnatural phrasing or awkward sentence structure.
- 4 – Clear and coherent; only minor language flaws.
- 5 – Highly fluent, polished, and well-suited for clinical or academic contexts.

**3. Caption/Dialogue Completeness:** Definition: For multi-turn dialogue, does the exchange include all key components of a meaningful clinical interaction (e.g., symptoms, history, reasoning, advice)? Evaluate whether the conversation flows logically and covers necessary content. For single-turn samples or caption, assess whether the response directly, sufficiently, and contextually addresses the input question or concern.

- 1 – Severely incomplete or off-topic; the response fails to address the input meaningfully.
- 2 – Major gaps; the response is only partially relevant or lacks necessary context.
- 3 – Generally appropriate, but missing some useful clarifications or elaboration.
- 4 – Mostly complete; clear and contextually suitable with minor detail omissions.
- 5 – Fully complete and coherent; the response provides an informative and context-aware answer, proportional to the input.

Note: For multi-turn dialogue, completeness includes aspects like logical progression, topic coverage, and closure. For single-turn Q&A, completeness means answering the question clearly, relevantly, and with appropriate medical insight.

**4. Medical Imaging Relevance:** Definition: If an image is present, does it clearly support or correspond to the associated text? Judge how well the image reinforces or illustrates the medical concepts being discussed.

---

- 1 – No image provided, or image is irrelevant/inappropriate. (Assign 1 by default if no image.)
- 2 – Weak connection; image adds little or may be confusing.
- 3 – Somewhat related; offers limited value or context.
- 4 – Relevant and supports the written content effectively.
- 5 – Strong alignment between image and text; image enhances understanding.

Note: If no image is provided in the sample, write: `"No image provided. Assigning a score of 1 by default."` and assign score = 1

**5. Practicality:** Definition: How useful is this data sample for real-world medical applications? Consider utility in model training, clinical decision support, educational value, or real patient interaction systems.
- 1 – No practical use; irrelevant or flawed content.
- 2 – Very limited applicability in specialized cases only.
- 3 – Somewhat useful; suitable for non-critical training or analysis.
- 4 – Practical and usable with minor improvements.
- 5 – Highly valuable for real-world use; clinically or technically actionable.

**Overall Score** Definition: Based on your evaluation across all five dimensions, assign a final overall score that reflects the holistic quality of the data sample. Consider accuracy, clarity, completeness, image relevance (if applicable), and practical usability as a whole. This score is not necessarily the average, but should represent your expert judgment of the sample's real-world value.
- 1 – Very poor overall; unreliable, misleading, or unusable.
- 2 – Weak quality; flawed in multiple aspects, limited usability.
- 3 – Adequate; some issues, but can be useful in certain contexts.
- 4 – Good quality; mostly solid with minor areas for improvement.
- 5 – Excellent; reliable, polished, and ready for real-world use or modeling.

The response fomat is:
```
{"Medical Information Accuracy": {"score": <1-5>,
"comment": "<explanation>"}, "Language Clarity and
Fluency": {"score": <1-5>, "comment": "<explanation>},
"Dialogue Completeness": {"score": <1-5>, "comment":
"<explanation>}, "Medical Imaging Relevance": {"score":
<1-5>, "comment": "<explanation>}, "Practicality":
{"score": <1-5>, "comment": "<explanation>"}, "Overall":
{ "score": <1-5>, "comment": "<summary comment>}}
```
Here is the sample: $\{s\}$

### A.3 DATA MIXING DETAILS

The training pipeline employs a structured four-stage data mixture strategy to progressively build the model's capabilities, as detailed in Table 4.

The process begins with continual pretraining on a large-scale foundation of both general-domain multimodal data (e.g., DataComp, OBELICS) and extensive medical caption data (e.g., Medtrinity-25M, ROCO), totaling approximately 30 billion tokens. This stage aims to establish robust visual and linguistic representations.

Next, the model's instruction-following ability is honed in two distinct phases. First, general instruction following is trained exclusively on the Mammoth-VL-10M dataset (∼10.1M samples). This is followed by medical instruction following, which combines a filtered portion of Mammoth-VL-10M with multiple medical instruction datasets (e.g., Path-VQA, PMC-VQA), resulting in a mixture of ∼11.69M samples.

Finally, for cross-distribution instruction adaption, the data is subsampled to create a balanced and high-quality mixture. This stage uses a small, curated set of ∼304k samples, comprising 180K from Mammoth-VL-10M and a balanced blend from key medical instruction datasets (e.g., 13K from LLaVA-Med-Instruct, 20K from PMC-VQA).

Table 4: The overview of the data mixture across the four training stages. Noted that `mammoth-VL-10M-filtered` variant excludes safety refusal responses (e.g., "Sorry, I can't..."), and `#number` denotes the number of samples after downsampling.

| Stage | Training Data Composition | Amount |
|---|---|---|
| Continual Pretraining | **1. General Multimodal Data** 
 DataComp, CCS, OBELICS, mmc4 
 **2. Medical Caption Data** 
 LLaVA-Med-60K-IM-Text, LLaVA-Med-Alignment, LLaVA-Med-Fig-Caption, Medtrinity-25M, 
 PubMedVision-Alignment, ROCO-radiology, ROCOv2-radiology | ∼ 30B tokens |
| General Instruction Following | Mammoth-VL-10M | ∼ 10.1M samples |
| Medical Instruction Following | **1. General Instruction Data** 
 Mammoth-VL-10M-filtered 
 **2. Medical Instruction Data** 
 LLaVA-Med-Instruct, Path-VQA, PMC-VQA, PubMedVision-Instruct Tuning, SLAKE, VQA-RAD | ∼11.69M samples |
| Cross-Distribution Instruction Adaptation | **1. Sampled General Instruction Data** 
 Mammoth-VL-10M-filtered#180K 
 **2. Sampled Medical Instruction Data** 
 LLaVA-Med-Instruct#13K, Path-VQA, PMC-VQA#20K, PubMedVision-Instruct Tuning#60K, SLAKE, VQA-RAD | ∼ 304K samples |

Table 5: Implementation details and hyperparameters for InfiMed-Foundation pretraining and supervised fine-tuning.

| Details | Pretraining | SFT |
|---|---|---|
| Vision Encoder | SigLIP-so400m-384px | SigLIP-so400m-384px |
| Visual Projector | Adaptive Average Pooling + MLP | MLP |
| LLM | Qwen3-1.7B-Instruct / Qwen3-4B-Instruct | Qwen3-1.7B-Instruct / Qwen3-4B-Instruct |
| Tokens per Image | 144 | 729 |
| Context Length | 4096 | 4096 |
| Sequence Packing | Yes | No |
| Global Batch Size | 256 | 128 |
| Training Epoch | 1 | 1 per stage |
| Optimizer | AdamW | AdamW |
| Peak LR | 5e-5 | 2e-5 |
| Warmup Ratio | 3% | 3% |
| Weight Decay | 0.01 | 0.01 |

## A.4 EVALUATION FRAMEWORK

To improve the efficiency and fairness of the evaluation, we adopt the MedEvalKit evaluation framework (Team et al., 2025b). This framework is designed to support a comprehensive set of mainstream medical benchmarks. MedEvalKit employs a standardized data preprocessing and postprocessing pipeline to ensure consistency and comparability of results. The framework implements a rule-based evaluation approach for closed-ended questions, which provides precise and objective scoring based on predefined criteria. For open-ended questions, MedEvalKit leverages an LLM-as-a-Judge strategy, utilizing advanced language models to assess response quality. Furthermore, the framework integrates vLLM (Kwon et al., 2023) for inference acceleration, significantly improving computational efficiency and scalability.

## A.5 IMPLEMENTATION DETAILS

The implementation details and hyperparameters for InfiMed-Foundation model pretraining and SFT are presented in Table 5. We used Qwen3-1.7B-Instruct and Qwen3-4B-Instruct as the large language model (LLM) backbones, and SigLIP as the vision transformer (ViT) backbone. The LLM input length is truncated at 4096 tokens. For pretraining, we utilized 32 NVIDIA H800 (80GB) GPUs. The optimizer is AdamW with a learning rate of 5e-5, weight decay of 0.01, and a warmup ratio of 0.03. The pretrain global batch size is 256. We pretrain the MLLM for one epoch. For supervised fine-tuning (SFT), we used 16 GPUs. The optimizer remains AdamW, but with a learning rate of 2e-5, a larger weight decay of 0.1, and the same warmup ratio of 0.03. The SFT global batch size is set to 128. Each stage of SFT is trained for one epoch.

## A.6 CASE STUDY

This section presents a series of case studies comparing the responses of Qwen2.5-VL-7B and our InfiMed-Foundation-4B model. Through qualitative analysis of specific examples, we demonstrate that InfiMed-Foundation-4B, despite its smaller size, exhibits a more solid grasp of medical knowledge, providing more accurate and reliable answers than its larger counterpart.

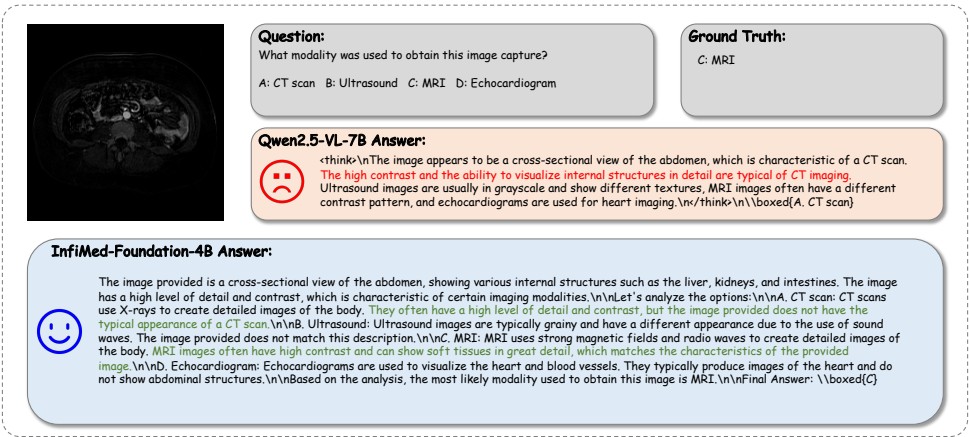

Figure 4: Case study No.1: Qwen2.5-VL-7B vs. InfiMed-Foundation-4B on Medical VQA.

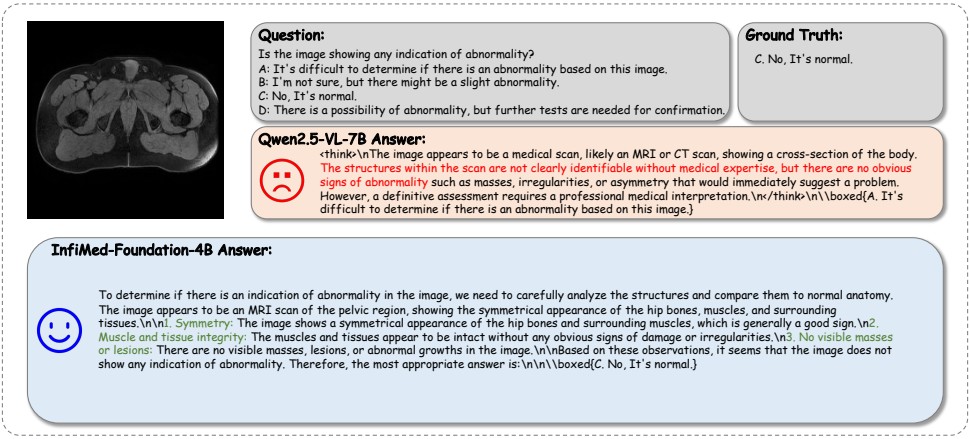

Figure 5: Case study No.2: Qwen2.5-VL-7B vs. InfiMed-Foundation-4B on Medical VQA. Qwen2.5-VL-7B was unable to make a judgment due to a lack of medical knowledge.

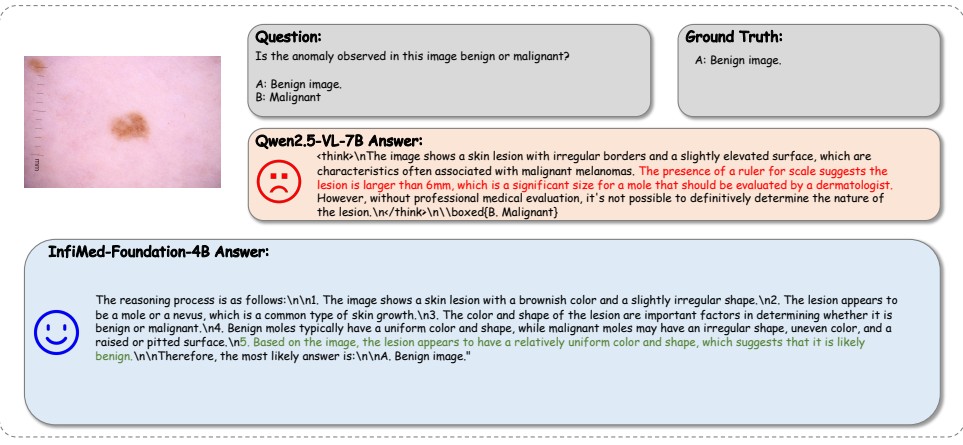

Figure 6: Case study No.3: Qwen2.5-VL-7B vs. InfiMed-Foundation-4B on Medical VQA. Qwen2.5-VL-7B made an incorrect estimation of the lesion size.

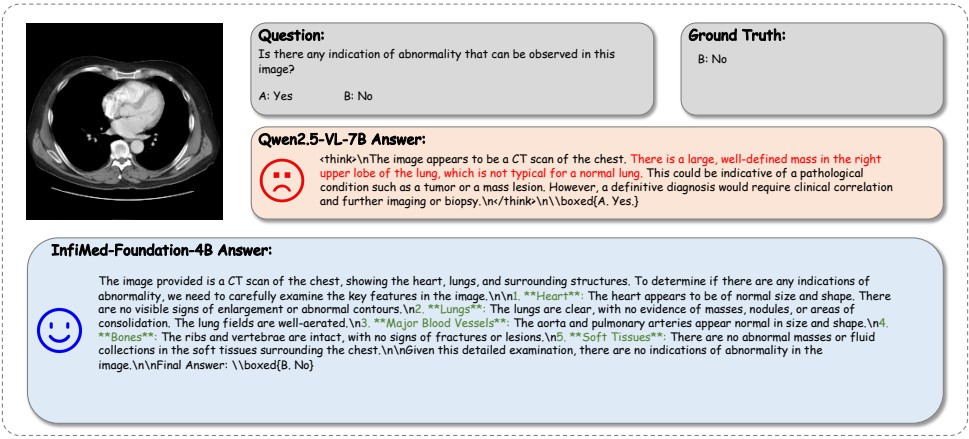

Figure 7: Case study No.4: Qwen2.5-VL-7B vs. InfiMed-Foundation-4B on Medical VQA. Qwen2.5-VL-7B produced an erroneous judgment, whereas InfiMed-Foundation-4B conducted a systematic analysis.

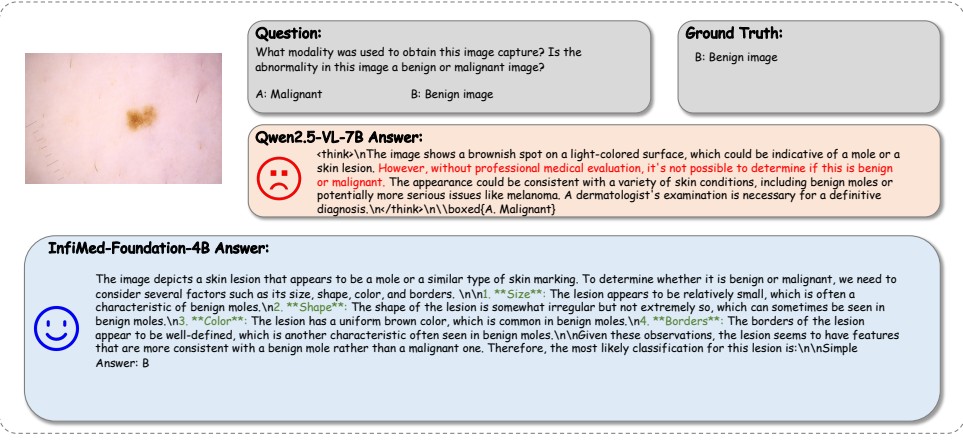

Figure 8: Case study No.5: Qwen2.5-VL-7B vs. InfiMed-Foundation-4B on Medical VQA. Qwen2.5-VL-7B exhibits a deficiency in medical knowledge.

