# OpenReview forum: "InfiMed-Foundation: Pioneering Advanced Multimodal Medical Models with Compute-Efficient Pre-Training and Multi-Stage Fine-Tuning"
_ICLR.cc/2026/Conference — Submitted to ICLR 2026_

### Official Review · Reviewer_P1Qy · 2025-10-30

**Soundness:** 2
**Presentation:** 2
**Contribution:** 1
**Rating:** 2
**Confidence:** 4

**Summary:**

This paper presents InfiMed-Foundation, a family of multimodal LLMs in the medical domain with a multi-stage supervised fine-tuning pipeline and compute-efficient pre-training. Using a five-dimensional quality framework, the authors curate a sizable medical multimodal dataset and integrate low-to-high-resolution training and multimodal sequence packing. MedEvalKit experiments demonstrate competitive performance when compared to open-source medical MLLMs.

**Strengths:**

1. A reasonably comprehensive engineering pipeline integrating existing data curation and training strategies.
2. Competitive results against some open-source baselines and modest compute-efficiency considerations.
3. Clear exposition and organized presentation.

**Weaknesses:**

1. The methodological novelty is limited. The proposed pipeline primarily recombines established techniques such as continual pretraining, staged SFT, etc. The overall contribution feels like a well-engineered extension of existing systems (e.g., LLaVA-Med, MedGemma, Lingshu, Open-Qwen2VL) rather than a conceptually new framework.
2. Although the proposed models demonstrate competitive performance, they do not achieve state-of-the-art results across all benchmarks. For end users, especially in medical AI applications, the primary selection criterion is performance rather than model scale or training cost, since most practitioners deploy pre-trained medical MLLMs rather than retraining them. As a result, despite training efficiency, the practical impact may be limited if users continue to choose higher-performing models (e.g., GPT-5 or Lingshu-7B). In other words, being parameter-efficient does not compensate for not being the top performer, and the model’s advantage may not translate into real-world adoption.
3. The performance gains on medical VQA tasks, while encouraging, do not clearly show superiority across diverse medical reasoning benchmarks. The paper does not include clinical report generation, temporal reasoning, radiology workflow tasks, medical conversation benchmarking, or structured knowledge grounding—limiting the claim of broad capability improvements.
4. The model remains weak on challenging tasks like MedXVQA (21.9%), and the evaluation focuses largely on VQA, without broader clinical tasks such as report generation or real-world clinical validation.

**Questions:**

See Weaknesses.

---

### Official Review · Reviewer_qjLD · 2025-10-31

**Soundness:** 3
**Presentation:** 3
**Contribution:** 2
**Rating:** 4
**Confidence:** 4

**Summary:**

This paper introduces InfiMed-Foundation-1.7B and InfiMed-Foundation-4B, two medical-specific multimodal large language models (MLLMs) designed to address challenges in applying general-purpose MLLMs to medical domains. The models are evaluated on the MedEvalKit framework, showing InfiMed-Foundation-4B outperforms comparable open-source models and approaches proprietary systems.

**Strengths:**

1. The paper presents a five-dimensional quality assessment framework for curating high-quality medical datasets.
2. The paper presents a compute-efficient pre-training strategy using multimodal sequence packing and reduced image patches.
3. The InfiMed-Foundation-4B outperforms comparable open-source models and approaches proprietary systems.

**Weaknesses:**

1. The proposed five-dimensional quality assessment framework lacks novelty and insight. Multiple prior works have already proposed multi-dimensional frameworks for evaluating data quality (e.g., data-centric AI literature, medical dataset curation papers). The authors fail to clearly articulate what is fundamentally new or superior about their specific five dimensions compared to existing frameworks. This contribution appears incremental rather than innovative.
2. Using adaptive average-pooling to reduce token count is well-established and not novel. More critically, the authors provide no comparison with state-of-the-art token reduction methods such as: Q-Former (BLIP-2)，Perceiver Resampler (Flamingo).
3. Table 2 shows InfiMed-Foundation's performance against open-source medical MLLMs is underwhelming: Significantly trails Lingshu-7B on multiple benchmarks (SLAKE: 77.7% vs 83.1%)

**Questions:**

1. The paper fails to establish a clear competitive advantage. Neither the technical innovations nor the model performance are sufficient for a high-impact publication at ICLR. The authors need to clearly articulate what fundamental contribution justifies acceptance at a top-tier venue.
2. Mixing general-domain data with medical data is not novel. The three-stage training pipeline closely resembles existing medical MLLMs, particularly Lingshu, without demonstrating clear technical differentiators or superior performance.

---

### Official Review · Reviewer_zgd5 · 2025-10-31

**Soundness:** 2
**Presentation:** 2
**Contribution:** 1
**Rating:** 2
**Confidence:** 4

**Summary:**

The authors introduce InfiMed-Foundation-4B and InfiMed-Foundation-1.7B, a family of multimodal LLMs trained for medical specialization. Starting from  The authors propose a multi-stage domain adaptation approach,

**Strengths:**

- The InfiMed models resulting from their training approach generally show good downstream performance on various medical VQA datasets, often comparable to or better than the performances of larger models (i.e., more parameters).
- The writing is generally well-written and easy to follow.

**Weaknesses:**

- The proposed multi-stage pretraining and fine-tuning approach (i.e., the training recipe) is not really novel. Modulo relatively minor differences, the multi-stage training approach is analogous to those initially proposed in prior works on visual instruction tuning [1,2,3] and medical adaptation of VLMs [4,5,6,7] (which also include approaches that mix general-domain and medical SFT datasets to mitigate catastrophic forgetting). In fact, many of the general-domain and medical adaptation datasets appear to be coming from these papers. As such, the contributions to the broader ML and ML for health communities appear limited.
- The paper frequently claims that the proposed approach is more computationally efficient, but there are no clear baselines to which their approach is being compared and no clear quantification or description of improvements in computational efficiency, and the results almost entirely focus on the downstream performance of the model. The most obvious efficiency gains appear to result from the sequence packing approach (to avoid wasting compute on padding tokens) and the the reduction in the number of image tokens, but neither approach are technically novel.
- It is not obvious how much impact the data quality evaluation step (Section 3.2) has on downstream performance. Ideally, the authors should quantify how much downstream performance improvement actually resulted from this filtering process via dataset ablations, although I understand that this can be computationally challenging. Nonetheless, I think it should be noted a limitation.
- Limitations of the study are not discussed.

References:
[1] Visual Instruction Tuning (Liu et al., 2023)
[2] Improved Baselines with Visual Instruction Tuning (Liu et al., 2024)
[3] InstructBLIP: Towards General-purpose Vision-Language Models with Instruction Tuning (Dai et al., 2023)
[4] LLaVA-Med: Training a Large Language-and-Vision Assistant for Biomedicine in One Day (Li et al., 2023)
[5] MediTron-70B: Scaling Medical Pretraining for Large Language Models (Chen et al., 2023)
[6] Med42-v2: A Suite of Clinical LLMs (Christophe et al., 2024)
[7] MedFlamingo: A Multimodal Few-shot Learner (Moor et al., 2023)

**Questions:**

Please see the Weaknesses section.

---

### Official Review · Reviewer_r88U · 2025-11-01

**Soundness:** 3
**Presentation:** 3
**Contribution:** 2
**Rating:** 4
**Confidence:** 5

**Summary:**

The paper introduces two medical-focused Large Vision-Language Models (LVLMs), InfiMed-Foundation-1.7B and InfiMed-Foundation-4B, with different model sizes. These models integrate high-quality open-source multimodal datasets from both general and medical domains, utilizing a five-dimensional quality assessment framework for data curation. To enhance training efficiency, the pretraining process adopts a progressive approach, starting with low-resolution images and increasing in resolution, along with multimodal sequence packing. The models are fine-tuned on three distinct data domains: general-purpose, medical-specific, and cross-domain data. Evaluations on the MedEvalKit framework show that InfiMed-Foundation-1.7B outperforms the Qwen2.5VL-3B model.

**Strengths:**

1. The paper effectively compiles and integrates available high-quality medical datasets, with the data selection process supported by a combination of LLM filtering and human evaluation, ensuring the quality of training data.

2. The paper is well-written, with clear and easily understandable explanations of the methods.

**Weaknesses:**

1. The paper lacks significant innovation; the main contributions are engineering-focused. The data preparation, CPT (Contrastive Pre-training), and SFT (Supervised Fine-Tuning) processes do not present new insights. Specifically:
a. Data: The data selection and filtering criteria follow common practices within the engineering community. The contribution mainly lies in the integration and filtering of existing open-source datasets.
b. CPT: The use of resolution adjustments and other training strategies has been common in pretraining smaller models for both general and medical domains. For instance, methods like BLIP[1] and MISS[2], which predate this work, have used different resolutions at various stages of training.
c. SFT: Stage-wise and mixed-data approaches have been commonly employed in recent research across various domains.

2. Benchmarking and Metrics: Compared to general-domain models, the reported metrics do not show a clear and substantial advantage. Furthermore, several of the benchmark datasets used, such as VQA-RAD and SLAKE, are outdated. For example, LLaVA-Med[3], a model evaluated on SLAKE, is a 2023 work, and the reported experimental results differ from the original papers (e.g., LLaVA-Med reports SLAKE metrics above 83%).
[1].Li J, Li D, Xiong C, et al. Blip: Bootstrapping language-image pre-training for unified vision-language understanding and generation[C]//International conference on machine learning. PMLR, 2022: 12888-12900.
[2].Chen J, Yang D, Jiang Y, et al. MISS: A generative pre-training and fine-tuning approach for med-VQA[C]//International Conference on Artificial Neural Networks. Cham: Springer Nature Switzerland, 2024: 299-313.
[3]. Li C, Wong C, Zhang S, et al. Llava-med: Training a large language-and-vision assistant for biomedicine in one day[J]. Advances in Neural Information Processing Systems, 2023, 36: 28541-28564.

**Questions:**

1.  The main contributions are engineering-focused.
2. Compared to general-domain models, the reported metrics do not show a clear and substantial advantage.

---

### Meta-Review · Area_Chair_HP1V · 2026-01-07

**Summary:**

The paper introduces two medical-focused Large Vision-Language Models (LVLMs), InfiMed-Foundation-1.7B and InfiMed-Foundation-4B, with different model sizes. These models integrate high-quality open-source multimodal datasets from both general and medical domains, utilizing a five-dimensional quality assessment framework for data curation. To enhance training efficiency, the pretraining process adopts a progressive approach, starting with low-resolution images and increasing in resolution, along with multimodal sequence packing. The models are fine-tuned on three distinct data domains: general-purpose, medical-specific, and cross-domain data. Evaluations on the MedEvalKit framework show that InfiMed-Foundation-1.7B outperforms the Qwen2.5VL-3B model.

There is no rebuttal submitted

**Reviewer Scores:**

N/A

---

### Decision · Program_Chairs · 2026-01-26

Reject